# Multiplexing Quantum and Classical Channels of a Quantum Key Distribution (QKD) System by Using the Attenuation Method

Ondrej Klicnik * , Petr Munster  and Tomas Horvath

Department of Telecommunications, Faculty of Electrical Engineering and Communication,
Brno University of Technology, 602 00 Brno, Czech Republic; munster@vut.cz (P.M.); horvath@vut.cz (T.H.)
* Correspondence: xklicn02@vutbr.cz

**Abstract:** The primary goal in this paper is to verify the possibility of combining a quantum channel into a single optical fiber with other classical channels by using the so-called attenuation method. Since the quantum channel is very weak in terms of power, combining it into a single fiber with much more powerful classical channels is challenging. Thus, sufficiently high-quality filtering is important to avoid possible crosstalk. A second and more difficult problem to address is the interference caused by Raman noise, which increases with the fiber length and is also dependent on the input power of the classical channel. Thus, in this paper the focus is on the possibility of suppressing the Raman noise effect, both in advance by means of wavelength positioning and by means of installed optical components. Such phenomena must be considered in the route design, as the quantum channel must be placed at a suitable wavelength with respect to the classical channels. The influence of other nonlinear phenomena has been neglected. In this paper, a practical experiment aimed at building a fully functional multiplexed quantum key distribution link is also described.

**Keywords:** attenuation; Clavis$^3$; coherent one-way protocol; quantum key distribution; wavelength-division multiplexing

## 1. Introduction

With the rapid development of quantum computing, the need for quantum-resistant key distribution is becoming increasingly urgent and important. In principle, there are two types of modern cryptographic techniques that should gradually replace the current asymmetric schemes based on factorization, discrete logarithm and elliptic curve problems.

One of these modern techniques is so-called postquantum cryptography (PQC). Similar to the algorithms currently in use, these algorithms are asymmetric but based on different mathematical problems, to which there are currently no known algorithms to break them even on a quantum computer. The alternative is quantum key distribution (QKD), which, in contrast, works based on the physical phenomena of quantum mechanics.

PQC techniques are fast and relatively straightforward to implement. They offer so-called computational security only. That is, their resistance to quantum computer attacks is not fully proven. In contrast, QKD offers so-called unconditional security. Thus, properly designed and implemented QKD protocols cannot be broken since breaking them corresponds to a violation of the rules of quantum mechanics.

However, in addition to increased security, the use of QKD in a real network produces a number of implementation challenges and high financial demands associated with high network resilience requirements as well as with the QKD devices themselves. One of the biggest challenges at the moment is the possibility of merging the quantum channel into a single fiber with classical channels. A more detailed description and differences between the two approaches can be found in [1].

In this paper, a method of coupling a weak quantum channel into a single fiber together with two much more powerful service channels is described. For the sake of

clarity, similar methods are referred to in the paper only as quantum multiplexing (QMUX). The service channels are emitted in opposite directions and jam the quantum channel with its Raman noise [2]. The basis of the entire setup is a research version of IDQ's Clavis[3] device with a key rate of 1400 b/s at a 14 dB attenuation. To make this setup work, the so-called attenuation method was chosen. In contrast to currently used techniques, the main advantage is the small spacing between quantum and classical channels. The techniques used are further elaborated in Section 3.

## 2. State of the Art

Although QKD is still considered by some to be a fringe issue belonging only to the arenas of intelligence communities and large enterprises, a growing number of new vendors are emerging, whether among universities or startups. Thus, QKD is currently being developed in many distinct directions, and the various systems are based on different physical principles and are at various stages of maturity. Although the vast majority of QKD transmissions are over optical fibers, many "wireless" versions are now also being actively developed. These are often referred to as free-space protocols. Among the most interesting are satellite systems such as the Chinese Micius project [3,4], which serves, among other things, as a link between Beijing and Vienna, and the European Space Agency's Eagle-1 project, where a satellite launch is planned for 2024 [5].

As in other areas of electrical engineering, there is a trend to minimize the individual components of a QKD system, which has led to multiple attempts to develop chip-based QKD and quantum random number generation (QRNG). An example may be Toshiba, which in 2021 introduced their first chip-based QRNG and QKD prototype [6]. Other organizations and startups, such as KETS, are developing their own solutions [7–9]. An alternative to most classical discrete-variable QKD (DV-QKD) protocols is the so-called continuous-variable QKD (CV-QKD), which stores information in multiphoton pulses by means of modulations. An example is the system implementation of the Technical University of Denmark (DTU) [10]. Although the coexistence of CV-QKD protocols with classical channels is not as well studied as DV-QKD, they are generally considered to be more robust to noise and thus more suitable for deployment in multiplexed networks [11,12]. Their second advantage is generally a higher key rate; however, their current disadvantage is usually a shorter range [13].

One of the largest obstacles of the recent QKD systems is the short maximum range of the quantum channel, which can, however, be extended in several ways. One is a well-known concept of a trusted repeater, which is a device that operates above the quantum layer itself and forward encrypted keys. In this way, it is possible to not only extend the link, but also build more complex topologies. However, a trusted node contains keys in nonquantum form; for this reason, it must itself be protected from access by an eavesdropper [14,15].

Various successes have also been achieved with different Measurement-device Independent QKD (MDI-QKD) protocols. Here, the most well-known representative is the twin-field QKD (TF-QKD). With this protocol, it is already possible to communicate over distances greater than 830 km under experimental conditions [16,17]. Recently, however, there have been reports of new protocols based on a similar principle that can outperform TF-QKD in terms of key rate [18,19].

In the near future, the so-called quantum repeaters might revolutionize the whole field of the quantum key distribution. Such repeaters are basically a new QKD protocol scheme based on the phenomenon of entanglement swap [20,21]. However, building them requires functional quantum memory that is not yet available. Therefore, quantum repeaters have not yet appeared. For this reason, quantum memories are currently one of the main areas of interest of many research labs, and there is gradual progress in this field [22,23].

## 3. Quantum Channel Positioning

As already mentioned, the quantum channel is very weak in terms of optical power. Typically, its power ranges between −80 and −90 dBm. Moreover, by the very nature of the transmission, it cannot be amplified or repeated (see the no-cloning theorem). Even in the case of dark fiber transmission, the maximum range of the QKD link is very limited. In practice, this is usually resolved by using so-called trusted repeaters, defined in ITU-T recommendation Y.3800. However, when additional channels are included, other malicious phenomena contribute to the degradation of the quantum channel. Among the most important ones are four-wave mixing (FWM) and spontaneous Raman scattering (SpRS/SRS). A detailed analysis of the impact of these and other phenomena on the quantum channel is discussed in articles [24,25].

While the FWM calculation is relatively straightforward and the interfering channel wavelength can be calculated by using the formula in [26], successful mitigation of the SpRS effect is much more challenging. The first step is to select the correct wavelength for all channels as described for Figure 1. This figure is based on the scattering characteristic measurements in papers [27,28]. Alternatively, it is possible to place the quantum channel far beyond the area of Raman scattering. However, even proper channel placement may not be sufficient in most cases. If the classical channel is powerful enough (or if there are multiple such channels), it is likely that one of the methods described below might be necessary to use.

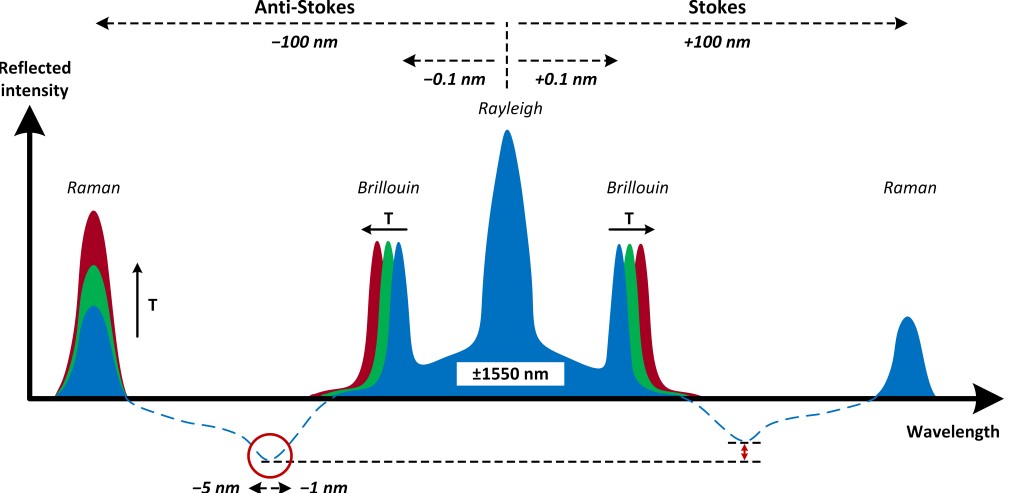

**Figure 1.** Sketch of a graph showing the approximate shape of the reflected noise of the classical channel in the band at approximately 1550 nm. In the middle is the central wavelength to which Rayleigh scattering corresponds. Then, at a distance of approximately 0.1 nm is the Brillouin scattering, and at a distance of 100 nm is the Raman scattering (both temperature dependent). In terms of noise, it is best to place the quantum channel in the anti-Stokes region 1 to 5 nm from the classical channel (red circle). Alternatively, a similar area might be used in the Stokes region.

## 4. Methods of Quantum Multiplexing

There are several ways to couple a quantum channel with others by using WDM, and all of them vary in efficiency and complexity. The main problem that these techniques address is the suppression of the influence of SpRS. The following overview contains the basic techniques known to the authors, but which may always need to be adapted to the desired application and combined with each other.

- **Large spacing method**—Commonly used by QKD system manufacturers. It is based on the largest possible spacing between quantum and classical channels (while respecting suitable wavelengths in the fiber). Typically, quantum channels are placed in the O-band (1310 nm), and classical channels are placed in the C-band (1550 nm). The total spacing is thus approximately 240 nm, and thus, Raman noise has minimal effect

on the quantum channel. Such a technique is simple to implement and is described with an example of a possible metropolitan network in [29].

- **Narrow filtering method**—Procedure described in [30]. Both types of channels are located in the C-band in relative proximity. Most of the Raman noise is filtered out by using the narrowest possible filters (DWDM 50 GHz, 25 GHz, 12.5 GHz).
- **Attenuation method**—Our method, which is described in detail in this article. Like the second one, the quantum and service channels are positioned close to each other. Here, however, conventional 100 GHz filters are used for filtering. For this reason, it is thus necessary to attenuate the other classical channels as much as possible. This also reduces the Raman noise power.

## 5. Attenuation Method Experiment

The essential part of this paper is the experimental verification of the QMUX attenuation method on a real commercial QKD system Clavis[3] and the evaluation of its behavior at various optical route distances. The system parameters are provided in Table 1. The measurements were carried out in three separate phases.

**Table 1.** Selected parameters of the system Clavis[3] used for the final route.

| Manufacturer | | | ID Quantique (IDQ) | | |
|---|---|---|---|---|---|
| QKD protocol | | | Coherent One-way (COW) | | |
| Pulse generation rate | | | 1.25 GHz | | |
| Key generation rate | | | 1.4 kbps | | |
| Dynamic range | | | 10–14 dB | | |
| Photon number | | | 0.03 | | |
| Quantum channel | $\longrightarrow$ | 1551.72 nm | CH32 | –83.2 dBm [1] | 1.4 kbps |
| Service channel I | $\longrightarrow$ | 1553.33 nm | CH30 | –3.0 dBm [2] | 2.7 Gbps |
| Service channel II | $\longleftarrow$ | 1554.13 nm | CH29 | –3.0 dBm [2] | 2.7 Gbps |

Arrows indicate the direction of the emitted light. [1] Based on own calculation. [2] Based on own power measurements.

1. **Design and safety verification of the initial route**—In this phase, a potentially functional QKD polygon with a route length of 20 km was designed. The length of the fiber was sufficient to manifest nonlinear effects while not exceeding the maximum possible quantum channel attenuation. DWDM filters were used to merge the quantum and classical channels, and their properties were tested by using an optical spectrum analyzer (OSA). Based on the results, these filters were then incorporated at a suitable location in the scheme. The optical power of the originally used Small Form-factor Pluggable (SFP) modules and the attenuation of all the optical elements used were also measured. Based on the calculations, it was verified that there could be no damage to any detection devices in the topology, especially the quantum signal detectors.
2. **Attenuation of service channels**—The second phase consisted of finding the minimum attenuation of the service channels so that the quantum channel was still functional. Testing was performed by using two pairs of SFPs. The first pair was used only for testing, i.e., no useful signal was sent, and the second pair provided a direct service channel link. This was the only way to ensure the correct operation of the entire system.
3. **Building the final route and tuning**—The final step was to build and test a fully functional QKD polygon. Unlike the previous phase, attention was already paid to the service channels, which are also a limiting element when sufficiently attenuated. For this reason, several modifications were made to the route. In particular, the replacement of lossy multiplexers by more attenuation-friendly circulators or the addition of an EDFA at the end of both service channels was carried out. The functionality of this route was tested for distances of 0, 5, 10, 15 and 20 km.

### 5.1. Phase I: Design and Safety Verification of the Iinitial Route

Within the first phase, a simple polygon was designed, as shown in Figure 2. In the case of neglecting the connections between the individual optical components, the total length was 20 km. This distance was chosen as an educated guess, as it is long enough to be affected by nonlinear effects. However, in terms of attenuation, it respects the manufacturer's specified dynamic range of 10–14 dB for the quantum channel.

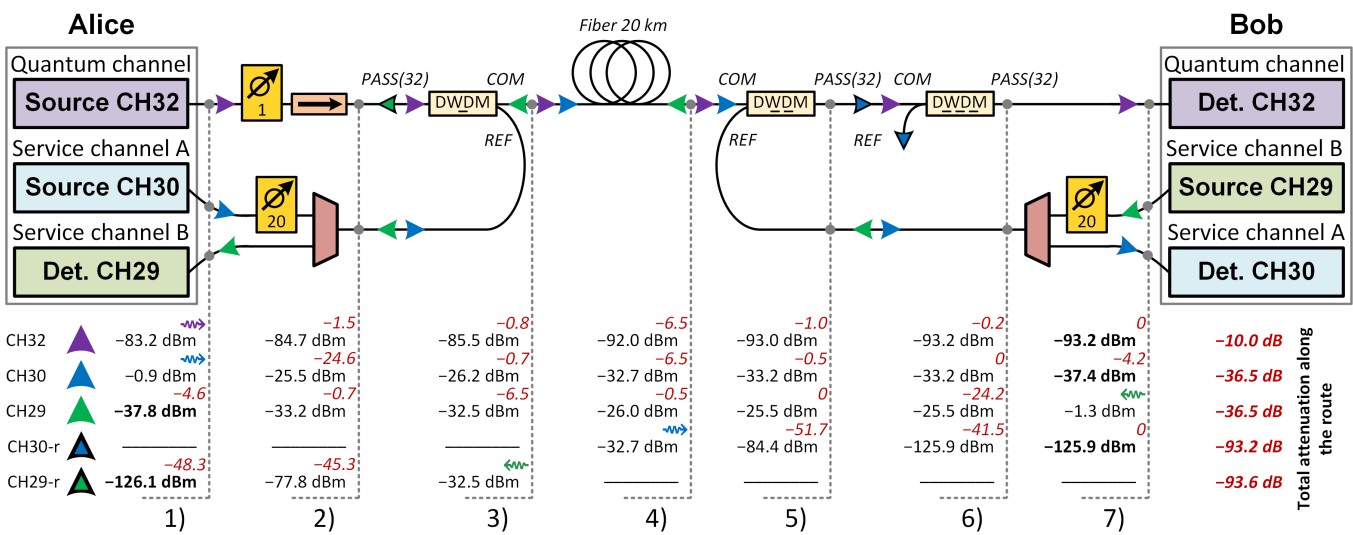

**Figure 2.** The original route combining two opposite service channels (CH29 and CH30) with one quantum channel (CH32) is connected. The layout also includes optical power values at important locations in the route, which were obtained by measurement or calculation. The values CH29-r and CH30-r represent the possible remaining power that was passed through the DWDM filters toward the source or detector of the quantum signal.

The presented scheme combines the quantum channel in one fiber with both of the service channels. The CH30 works in the same direction as the quantum channel, i.e., is transmitted from Alice toward Bob, and the direction of the second service channel is the opposite. To properly filter out noise from the quantum channel, it is necessary not only to properly filter out extraneous wavelengths, but also to ensure that noise in the area of CH32 does not reach the shared fiber. Such noise would subsequently no longer be distinguishable from the quantum channel.

Both service channels are first combined into a single fiber by using a conventional multiplexer. However, in addition to its main role, the MUX serves as a filter that removes most of the noise on CH32. To keep the quantum channel interference to a minimum, the service channel is attenuated by 20 dB at the source.

The basis of the topology is three DWDM filters transmitting CH32, for which qualitative measuring of parameters was conducted. Based on their measured characteristics, they were placed in Positions I, II and III from left to right (marked with underscores).

- **I. Position**—The main function of the filter is to bind CH30 to a common fiber with a quantum channel and simultaneously extract CH29 from the fiber. Its secondary function is similar to that of the multiplexers described above, that is, to filter out the noise produced by CH30 in the CH32 region. Most of the noise that passes through the MUX is passed to the PASS port, where it is then absorbed by the isolator along with the remnants of CH29.

- **II. Position**—As in the previous case, the main function of the filter is to bind and unbind the service channels to a common fiber. In this case, however, it is no longer appropriate for the filter to pass the noise of service channel B. Unlike Filter I, most of the noise in the CH32 region must be reflected toward Alice, where it again passes to the isolator and is absorbed.

- **III. Position**—The filter is mainly used to increase the OSNR (optical signal-to-noise ratio) of the quantum channel. Therefore, its isolation in the COM-PASS direction is important. Service channel residues that are not suppressed by the previous filter are removed from the fiber by the REF port.

There are two more optical elements on the quantum channel: an isolator, which prevents any light from passing through the filter from reaching the source, and a 1 dB attenuator, which increases the attenuation on the quantum channel to a total dynamic range of 10 dB.

*5.2. Phase II: Attenuation of Service Channels*

The following measurement that is shown in Figure 3 directly follows the previous phase and is aimed at finding the minimum possible attenuation of the service channels so that they do not interfere with the quantum channel after coupling, so that the system remains functional. However, unlike the first phase, this measurement requires a working QKD system. To evaluate something like this, it is necessary to connect the service channels directly so that they do not interfere with the quantum channel and at the same time their synchronization is not disturbed when they are attenuated. For this purpose, a second pair of SFP modules must be used.

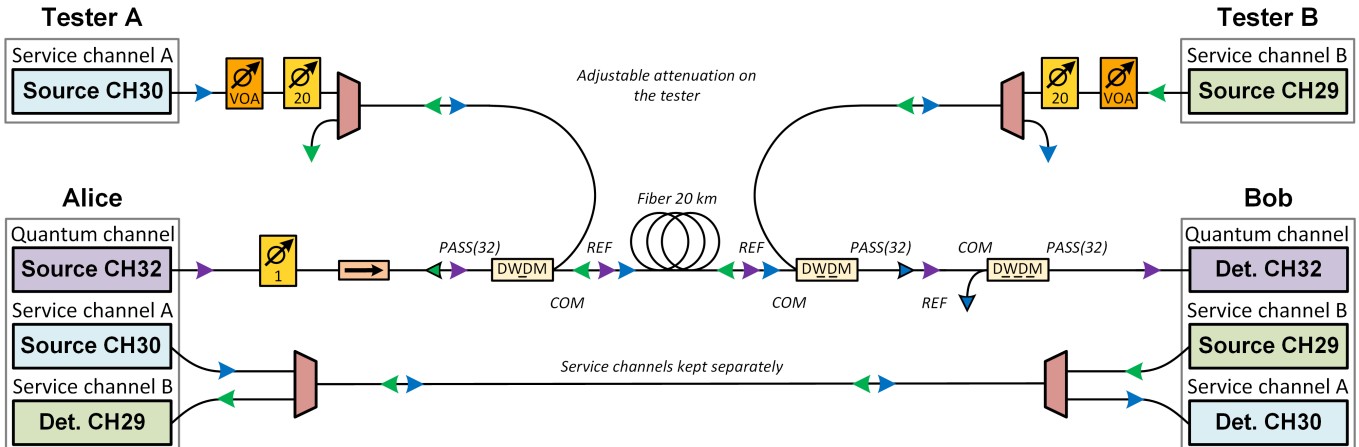

**Figure 3.** Initial route from Phase I, modified for service channel attenuation testing. The main difference is the addition of a second SFP pair.

The other pair of SFPs is plugged into an arbitrary device and is used solely for interference testing. Thus, it does not transmit any useful signal and can be attenuated at will by using a variable optical attenuator (VOA). This adds additional attenuation to the 20 dB already designed. Since the quantum channel is not functional at a 20 dB attenuation, there is no point in reducing the attenuation.

As in the previous phase, the transmitted power is monitored. The measurement principle itself consists of gradually adding attenuation to the VOA and monitoring changes in the QKD system parameters. In total, testing is performed for both service channels used simultaneously in 10 steps. Subsequently, the values of the key rate, visibility and quantum-bit error rate (QBER) are obtained. Their averages for the individual attenuation values can be found in Table 2.

Table 2 contains the quantum channel performance values for both connected service channels simultaneously. The results confirm the expectation that as the value of service channel attenuation decreases, the QBER increases, and visibility decreases with the transmission rate. Thus, the minimum route configuration should contain at least 5 dB of added attenuation.

**Table 2.** Quantum channel performance when combined with both service channels. The first column contains the attenuation added by the VOA to the already installed 20 dB.

| SFP 29 and 30 | Key Rate (kbps) | QBER (%) | Visibility (%) |
|---|---|---|---|
| Turned off | 2.07 | 3.04 | 98.60 |
| +20 dB | 2.04 | 3.22 | 98.40 |
| +10 dB | 1.59 | 3.41 | 97.70 |
| +7 dB | 1.26 | 4.10 | 97.20 |
| +5 dB | 0.65 | 4.45 | 97.40 |
| +4 dB | non-functional | 5.05 | 97.10 |
| +3 dB | non-functional | 5.27 | 96.70 |
| +2 dB | non-functional | 5.32 | 96.50 |
| +1 dB | non-functional | 6.56 | 96.00 |
| +0 dB | non-functional | 8.96 | 94.80 |

5.2.1. Crosstalk or Raman Noise?

To determine the reason for the decreasing key rate, a quick test was performed as part of the phase. From the configurations in Table 2, the one with a +7 dB attenuation added was selected. This is a configuration where the system is in a stable state and at the same time a change in Visibility, Key Rate and QBER can be observed.

Subsequently, the 20 km fiber was removed, and Filters I and II were connected directly. By bending the interconnecting cable of negligible length between the filters, the same attenuation was added as the originally connected route (approximately 6 dB).

If the quantum channel interference is caused by crosstalk, its value no longer increases with channel length, and the system parameters remain the same in both cases. In contrast, Raman noise increases rapidly with channel length. Thus, from the results shown in Table 3, Raman noise is the main cause of quantum channel interference. The possible contributions of other nonlinear phenomena are neglected.

**Table 3.** Difference in quantum channel performance depending on the route length.

| Connection | Key Rate (kbps) | QBER (%) | Visibility (%) |
|---|---|---|---|
| Direct | 2.11 | 3.33 | 98.70 |
| 20 km | 1.21 | 4.05 | 97.40 |

5.2.2. Raman Noise Estimation

Raman noise is therefore the main limiting element of the whole scheme; for this reason, it is appropriate to calculate its approximate optical power. Based on the formulas from [27] given below, its values for the smallest (+0 dB) and largest (+20 dB) used service channel attenuations were calculated.

- Raman noise power generated in the forward direction ($\rightarrow$):

$$P_{\overrightarrow{SpRS}} = P_{in} \cdot e^{-A} \cdot \rho(\lambda) \cdot L \cdot \Delta\lambda. \tag{1}$$

- Raman noise power generated in the backward direction ($\leftarrow$):

$$P_{\overleftarrow{SpRS}} = P_{in} \cdot e^{-A} \cdot \rho(\lambda) \cdot \frac{sinh(L\alpha)}{\alpha} \cdot \Delta\lambda. \tag{2}$$

Both service channels contribute to Raman noise, which is generated in both forward and backward directions. Since the two channels emit in different directions, only forward Raman noise of CH30 and backward Raman noise of CH29 are incident on the QKD detector. The total malicious Raman noise power is then determined as the sum of these two outputs. The illustration can be found in Figure 4 below.

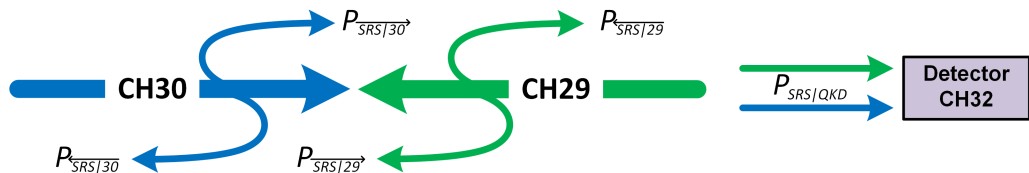

**Figure 4.** The figure shows what components of Raman noise produce errors in detecting a quantum signal. The total malicious Raman noise consists of the forward component CH30 and the backward component CH29.

Since Raman scattering is a nonlinear phenomenon, the calculation must be made in basic units (not decibels). Parameters of the following values are used for the calculation with Formulas (1) and (2):

- **Output power**—$P_{out}$—optical power of the service channel (both have the same value) entering the first DWDM filter. This power must be converted to mW. It can be calculated as $P_{out} = P_{in} \cdot e^{-A} = P_{in} \cdot e^{-\alpha \cdot L}$, where $A$ denotes the attenuation.

$$P_{out|+0 \text{ dB}} = -31.5 \text{ dBm} = 7.08 \cdot 10^{-4} \text{ mW},$$

$$P_{out|+20 \text{ dB}} = -51.5 \text{ dBm} = 7.08 \cdot 10^{-6} \text{ mW}.$$

- **Raman scattering coefficient**—$\rho(\lambda)$—determined from the Raman cross-section graph in [27] for channels two and three positions away from the quantum channel in the Stokes scattering region.

$$\rho(\lambda) = 1.85 \cdot 10^{-9} \frac{1}{\text{km} \cdot \text{nm}}.$$

- **Bandwidth**—$\Delta\lambda$—the region of the spectrum in which the Raman scattering power is measured. The DWDM channel has a width of 100 GHz, which must be converted to nanometers.

$$\Delta\lambda = 0.8 \text{ nm} \approx \Delta\nu = 100 \text{ GHz}.$$

- **Route length**—$L$—the part of the route where Raman scattering and other nonlinear effects occur.

$$L = 20 \text{ km}.$$

- **Attenuation coefficient**—$\alpha$—calculated based on the difference between input and output optical power, and then converted to $\text{km}^{-1}$.

$$\alpha = 0.3 \frac{\text{dB}}{\text{km}} = 6.91 \cdot 10^{-2} \text{ km}^{-1}.$$

Based on these values and formulas, both malicious components and their sum can then be calculated, thus expressing the total malicious Raman noise power that passes through the 100 GHz DWDM filter to the single-photon QKD detector. These results limit the range of SpRS optical power.

- **Minimum attenuation (+0 dB)**
  - $P_{\overrightarrow{SpRS|30}} = -106.79$ dBm,
  - $P_{\overleftarrow{SpRS|29}} = -105.48$ dBm,
  - $P_{SpRS|QKD} = -103.08$ dBm.

- **Maximum attenuation (+20 dB)**
  - $P_{\overrightarrow{SpRS|30}} = -126.79$ dBm,
  - $P_{\overleftarrow{SpRS|29}} = -125.48$ dBm,
  - $P_{SpRS|QKD} = -123.08$ dBm.

As expected, in both cases, the Raman backscatter is slightly higher than the forward scatter. Overall, however, the difference is minimal and does not change much with increasing attenuation. At this distance, the difference can thus be neglected, and the power of the malicious Raman noise in both directions can be declared to be equal. By summing them together, the power obtained is twice as high (+3 dB).

*5.3. Phase III: Building the Final Route and Tuning*

The final step of this paper is to build a final and functional optical polygon fully utilizing a shared fiber to carry the quantum and both service channels. Unlike the previous step, the functionality of the service channels must now also be considered, as their detectors also have only limited sensitivity. For this reason, several changes are made compared to the previous setup, as seen in diagram Figure 5. While most of them are related to service channels, only one is related to the quantum channel. This is the relocation of the attenuator closer to the detector. Thus, not only Channel 32 but also the noise is attenuated. Such a solution ensures a higher OSNR.

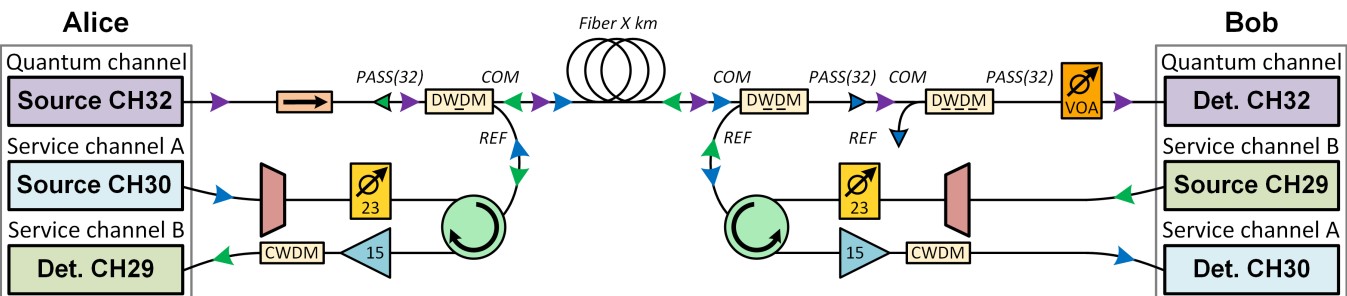

**Figure 5.** Schematic of the final and functional route of the polygon. Compared to the previous design, it contains a number of different components. In particular, these are circulators and EDFAs whose signal is subsequently "trimmed" by using a CWDM filter.

As already indicated, the limiting element is no longer just the quantum channel. For proper functioning, it is necessary to attenuate the service channels in such a way that their resulting optical power is far below the desired sensitivity of the detector. For this reason, EDFA and lower attenuation components must be used, as shown in Figure 6.

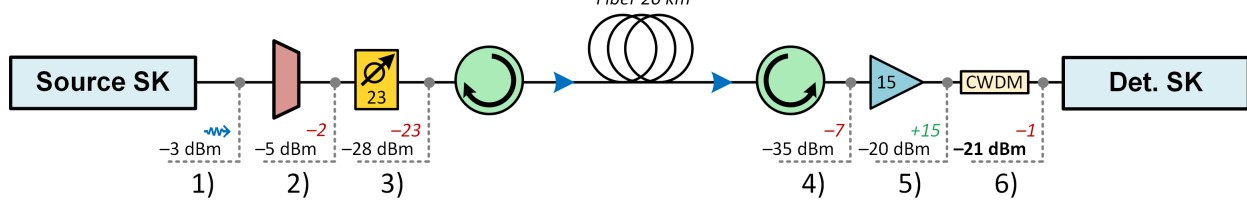

**Figure 6.** Schematic diagram of one of the service channels with power values at various points along the optical route.

The first change is the deployment of new SFPs with a more stable optical power value (−3 dBm). This allows us the removal of part of the added attenuation. Both channels are measured, and their spectra can be found in plot Figure 7A. The quantum channel region is also marked in purple for comparison.

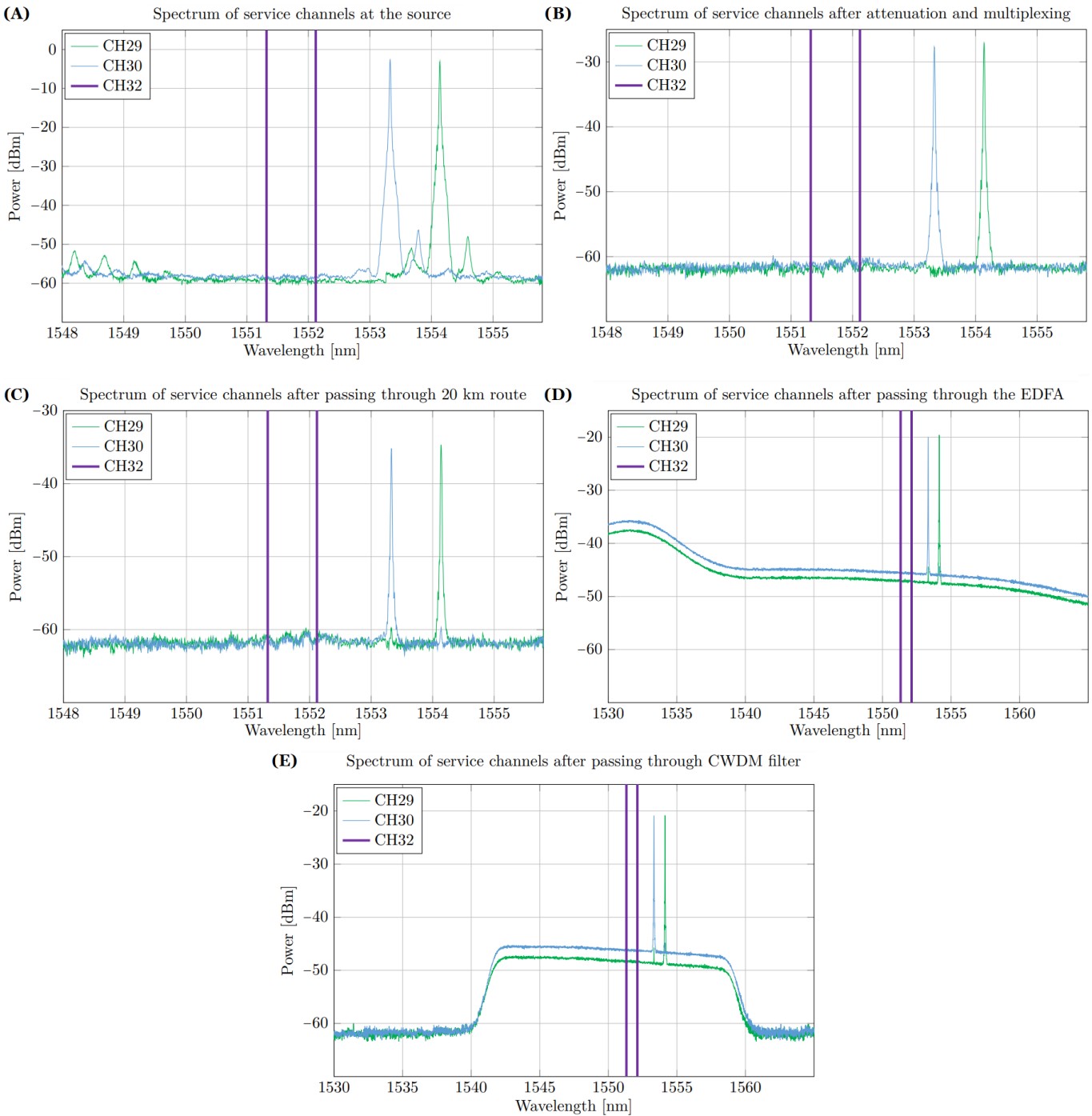

**Figure 7.** Graphs showing the spectra of both service channels (CH29 and CH30) in different parts of the optical route. The purple lines show the region of occurrence of the quantum channel (CH32).

In the original topology, the service channels are first multiplexed into a single fiber by using multiplexers with an attenuation of approximately 4.5 dB. In the same way, they are separated on the opposite side. However, this arrangement adds high attenuation to the path, which is not problematic in the section before the connection to the shared path where the service channels are intentionally attenuated. However, once the shared section is passed, the power of the service channels must be at least equal to the sensitivity of the detectors. For this reason, further attenuation is not desired. The two channels are opposite to each other, which allows replacement of the multiplexers by circulators with much lower attenuation (approximately 0.5 dB).

Removing multiplexers, however, leads to noise in the quantum channel. Although the service channels transmit on adjacent channels, they produce noise in the CH32 region in addition to the useful signal. This must be filtered out before all channels are combined. For this reason, a multiplexer is connected to the start of the route, but it only serves as a filter. By reconfiguring the attenuator, the same input power is obtained as in the previous stage. The spectrum of service channels after multiplexer "trimming" is shown in Figure 7B.

As shown in Figure 7C, after passing through the shared path and the circulators (total attenuation of approximately 7 dB), the power of both service channels is much lower (−35 dBm) than the sensitivity of the SFP modules used (−24 dBm). For this reason, an EDFA is added on each side.

The spectra of both channels after passing through the amplifier are located in Figure 7D. This shows that the EDFA amplifies most in the region at approximately 1530 nm. Since the detector accepts wavelengths in the range of approximately 1528–1569 nm, there would be too much error due to the small signal-to-noise ratio. For this reason, a CWDM filter operating at $\lambda$ = 1550 nm is placed between the amplifier and the detector itself. The resulting signal therefore appears in Figure 7E.

Reducing the Route Length

Although the sensitivity of the used SFP detectors is approximately −24 dBm, the transmitted signal is so damaged by the strong attenuation and subsequent amplification that the system needs to deliver at least −21 dBm for proper operation. Based on the above and Figure 8, it appears that this is coincidentally the same power obtained with the minimum attenuation needed for the quantum channel to operate. This issue is explained in Figure 8.

**Service channel power**

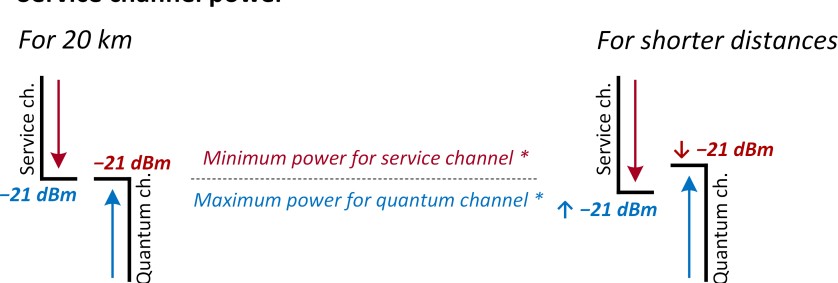

\* *Minimum/maximum service channel power for proper service/quantum channel operation*

**Figure 8.** The graphic describes the service channel optical power problem. The higher its power, the worse the effect it has on the quantum channel. In contrast, the more the service channel is attenuated, the lower the probability that it works properly. The "benevolence" in terms of power increases as the shared fiber shortens.

However, it is necessary to add that this is a limit state in which keys are indeed transferred but only at a very low rate and for a limited time. The system is unstable, and after a certain period of time, the whole key distribution process restarts. However, as the distance decreases (attenuation along the path is compensated), the quantum channel is less disturbed by Raman noise and is thus able to operate even at higher service channel optical power. Moreover, if the power of the service channels is preserved, this leads to higher system stability and consequently higher transmission rates.

The route was therefore shortened four times with a step of 5 km until both DWDM filters were connected directly. The path attenuation was always adjusted by bending to the original value (i.e., of approximately 6 dB). The attenuation of the service channels as well as their final performance was thus maintained. The differences in key rate, error rate and visibility can thus only be attributed to Raman noise. Calculation of its optical power was performed in the same way as in the Section 5.2.

## 6. Discussion of the Results

Although the 20 km length of the shared fiber has an impact on the Key Rate and stability of the system, it can be concluded that transferring quantum keys over a shared fiber over such distance is possible. The same measurements were made for shorter distances, but at which the original attenuation was preserved.

The results in Table 4 show that the key rate of the system increases with decreasing distance. Thus, the transmission is less disturbed by Raman noise, as evidenced by the decreasing QBER and increasing visibility. Therefore, the last phase can also be considered evidence that nonlinear phenomena, particularly Raman noise, have a significant effect on the quantum channel.

**Table 4.** Measurement results for five QKD configuration of the polygon. The table contains the Raman noise optical power values and basic system parameters.

| Connection | Raman (dBm) | Key Rate (kbit/s) | QBER (%) | Visibility (%) |
|---|---|---|---|---|
| Direct | —— | 2.52 | 2.78 | 99.30 |
| 5 km | −112.75 | 2.19 | 3.57 | 98.00 |
| 10 km | −109.61 | 2.04 | 3.67 | 97.50 |
| 15 km | −107.63 | 1.69 | 3.92 | 97.50 |
| 20 km | −106.08 | 0.45 | 4.61 | 97.30 |

It is possible that with the use of better optical components, the route can be extended further. The prerequisites are, e.g., more sensitive SFP detectors and more powerful EDFA, which may eventually allow more attenuation of the service channel. A second suggested option is combination with the narrow filtering method to filter out more Raman noise. The experiment presented here used the location of the service channels at the lowest point of the Stokes noise region. However, placing the service channels in the anti-Stokes region should further reduce the overall power of malicious Raman noise.

## 7. Conclusions

In this paper, the capabilities of QMUX were discussed, and the techniques known to date were outlined. The appropriate placement of the quantum channel with respect to Raman noise was also discussed. The main part was the introduction of the attenuation method, the functionality of which was experimentally tested. The experiment consisted of three successive phases. First, a prototype QKD polygon was designed, and the safety of all optical elements was tested. In the next phase, a service channel power limit was determined by means of successive attenuation. Based on this measurement, additional optical elements were added to the scheme in the third phase. Thus, a functional polygon with a variable length of the shared route was created.

**Author Contributions:** Conceptualization, O.K.; validation, T.H. and P.M.; writing—original draft preparation, O.K.; visualization, O.K.; supervision, P.M.; project administration, T.H. All authors have read and agreed to the published version of the manuscript.

**Funding:** This work is supported by the Ministry of the Interior of the Czech Republic, program IMPAKT1, under grant VJ01010008, project Network Cybersecurity in the Post-Quantum Era.

**Data Availability Statement:** The data presented in this study are available on request from the corresponding author. The data are not publicly available.

**Conflicts of Interest:** The authors declare no conflict of interest.

## Abbreviations

The following abbreviations are used in this manuscript:

| | |
|---|---|
| COW | Coherent One-way |
| CV-QKD | Continuous-variable QKD |
| CWDM | Coarse WDM |
| DTU | Technical University of Denmark |
| DV-QKD | Discrete-variable QKD |
| DWDM | Dense WDM |
| EDFA | Erbium-Doped Fiber Amplifier |
| FWM | Four-Wave Mixing |
| IDQ | ID Quantique |
| ITU-T | International Telecommunication Union—Telecommunication Standardization Sector |
| MDI-QKD | Measurement-device Independent QKD |
| MUX | Multiplexer |
| OSA | Optical Spectrum Analyzer |
| OSNR | Optical Signal-to-noise Ratio |
| PQC | Postquantum Cryptography |
| QBER | Quantum-bit Error Rate |
| QKD | Quantum Key Distribution |
| QMUX | Quantum Multiplexing |
| QRNG | Quantum Random Number Generation |
| SFP | Small Form-factor Pluggable |
| SpRS | Spontaneous Raman Scattering |
| TF-QKD | Twin-field QKD |
| VOA | Variable Optical Attenuator |
| WDM | Wavelength-division Multiplexing |

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
