# Peer review of "Multiplexing Quantum and Classical Channels of a Quantum Key Distribution (QKD) System by Using the Attenuation Method"

_photonics, doi:10.3390/photonics10111265_

Round 1
Reviewer 1 Report
Comments and Suggestions for Authors
The purpose of the article of Ondrej Klicnik, Petr Munster and Tomas Horvath "Multiplexing Quantum and Classical Channels Using the Attenuation Method" is to investigate the possibility of combining a quantum channel in one optical fiber with other classical channels and to justify the suppression of the Raman noise effect.
The topic of the article is in the trend of the most modern research on quantum optics. The article is certainly of interest to the readers of the journal.
Of course, one can always find some flaws, but I liked this article. It is not only in the trend of modern research and written in clear language, but also has obvious applied potential. I believe that the article can be published as presented.
Reviewer 2 Report
Comments and Suggestions for Authors
The proposed research's efforts and contributions to the field of QKD are fully acknowledged and recognized as an important research topic. However, it appears that the overall paper structure and description method will need to be revised as described below.
1. The title should be described in more detail to clarify the characteristics of the manuscript.
2. Additional explanation about QMUX-related research is needed in Section 1.
3. It consists of a total of 10 sections, which reduces readability, so it is recommended to reorganize it into 5 to 6 sections.
4. It is recommended to add more related references.
Comments on the Quality of English LanguageMinor editing of English language required
Reviewer 3 Report
Comments and Suggestions for Authors
Multiplexing quantum and classical channels has been a significant area of research in the field of quantum key distribution. This paper investigates the feasibility of combining a quantum channel with other classical channels into a single optical fiber using the attenuation method. The focus of this study is on the possibility of suppressing Raman noise effects in advance through wavelength positioning and the installation of optical components. However, there are some elements that are still unclear and need to be modified. The manuscript should be modified and improved. My suggestions are as follows (see list below).
1. In the Introduction section, quantum key distribution has been mentioned to establish the research focus. However, there is a lack of a detailed description of the relationship between multiplexing quantum and classical channels and quantum key distribution. It is advisable to include more relevant content and cite additional articles to provide a deeper background and context for the research.
2. In line 113, "the previous one" could be misleading. Using "the second one" or specifying "the narrow filtering method" would enhance the readability of your article.
3. In Figure 1, the meaning of the red circle and red double arrow is not clear. Further clarification and labeling would be beneficial. Additionally, you can include symbols or labels to represent positions I, II, and III in the DWDM.
4. In lines 159 and 166, replace "channel 32" with "CH32" to maintain consistency and clarity.
5. In equation (1),
can be written as
(Att) (where Att is the attenuation value, a suitable sign should be chosen to represent it) to show the influence of attenuation on Raman noise power.
6. The multiplexing technique plays an important role in quantum key distribution and quantum cryptography. Adding more theoretical new protocols [PRX Quantum 3, 020315 (2022); Nature 557, 400 (2018)] and experimental works [Nat. Photonics 8, 595–604 (2014); Phys. Rev. Lett. 130, 250801 (2023)] can further highlight the practical implications and potential applications of the proposed work. Therefore, I suggest that the authors should cite the above articles which can make your article be better and make the reader have more background.
Once the authors have addressed the points raised, I will agree that this work can be published in Photonics.
Comments on the Quality of English LanguageModerate editing of English language required
Round 2
Reviewer 3 Report
Comments and Suggestions for Authors
This manuscript can be accepted for publication.